# Circulating Tumor DNA in Oncology

Saeko Sakaeda [1] and Yoichi Naito [2,*]

1 Department of Medical Oncology, National Cancer Center Hospital East, 6-5-1 Kashiwanoha, Kashiwa 277-8577, Chiba, Japan; ss6584@tokai.ac.jp
2 Department of General Internal Medicine, National Cancer Center Hospital East, 6-5-1 Kashiwanoha, Kashiwa 277-8577, Chiba, Japan
* Correspondence: ynaito@east.ncc.go.jp; Tel.: +81-4-7133-1111; Fax: +81-4-7131-4724

**Abstract:** When somatic cells in the human body undergo apoptosis or necrosis, the released DNA enters the bloodstream. This type of DNA is called cell-free DNA (cfDNA). In patients with cancer, DNA released from tumor cells is called circulating tumor DNA (ctDNA), which carries genetic alterations specific to tumor cells. In recent years, ctDNA has attracted particular attention in terms of the concept of liquid biopsy in cancer care. Conventionally, tissue biopsy is required for the definitive diagnosis of cancer, and imaging examinations, such as CT, are performed for evaluating recurrence and residual lesions. Although the treatment burden on cancer patients is being slightly reduced due to advances in medicine, invasive examinations and medical exposure are still unavoidable. In addition, the prognosis of cancer varies considerably depending on the degree of progression at the time of detection. Therefore, the early detection of cancer is of utmost importance. With the increase in health consciousness, more people undergo regular health checkups, and it becomes necessary to diagnose cancer in a larger number of patients at an earlier stage. Although the accuracy of early detection has been improved by new imaging tests and examination techniques, each organ must be examined separately, and some organs are more difficult to examine than others in a regular health checkup. The process of cancer screening, diagnosis, and detection of recurrence after treatment is extensive. It can also be expensive, and some of the examinations may be invasive. If all of these processes can be replaced by the analysis of ctDNA in liquid biopsy, only a single blood sample is required. Under these circumstances, various studies are currently in progress on the use of ctDNA in clinical practice as an approach that may greatly reduce such burden. We present an overview of the current situation of ctDNA, as well as its future issues and prospects.

**Keywords:** ctDNA; liquid biopsy; cancer screening

## 1. Introduction

The first report on cfDNA in human blood was made by Mandel and Métais in 1948 [1]. Then, in 1998, the presence of DNA originating from the fetus in pregnant women's blood was discovered [2]. Since then, the application of cfDNA to prenatal fetal diagnosis has rapidly spread in clinical practice [3,4].

In 1989, Stroun, M., et al. discovered a cfDNA fraction derived from cancer among fractions of cfDNA in the plasma of cancer patients. This was the first report of ctDNA [5,6]. In 1994, the *KRAS* mutation was identified in plasma cfDNA from pancreatic cancer patients by the PCR method. As this mutation was identical to the *KRAS* mutation detected in the tumor tissue from the same patients [7], it was confirmed that the DNA mutation in the plasma was tumor-derived, and the term "circulating tumor DNA (ctDNA)" was established.

Cell-free DNA is released into the bloodstream through cell apoptosis or necrosis and can also be detected in healthy individuals. It has been reported that plasma cfDNA concentrations in healthy individuals are often in the range of 1–10 ng/mL [8,9]. In addition,

the concentration of cfDNA increases not only in patients with cancer, but also in those with infections, trauma, and cerebrovascular diseases [10].

Liquid biopsy is broadly defined as a minimally invasive examination to detect ctDNA (Figure 1), circulating tumor cells (CTC), miRNA, etc. in body fluids [11–13]. We focus on ctDNA in this review.

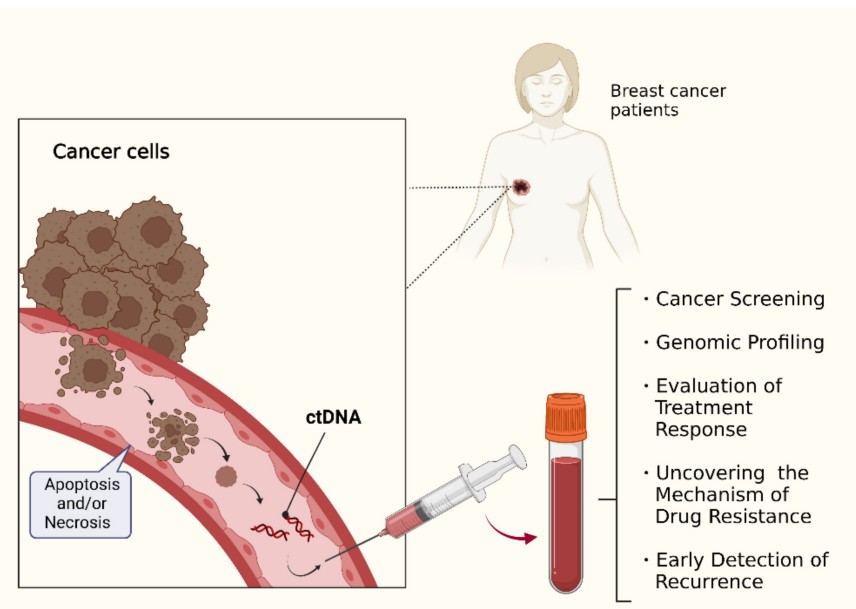

**Figure 1.** ctDNA and its utility. ctDNA is expected to be useful in various situations of cancer care including diagnosis, treatment, and recurrence. (Created with BioRender.com, 31 October 2021).

Currently, a biopsy of the primary or metastatic lesion is performed in cancer diagnosis, and a definitive diagnosis is made using pathological diagnosis. The process, up to diagnosis, involves frequent medical exposure during diagnostic imaging and patients' physical distress associated with biopsy. The larger the tumor, the easier from which it is to collect a tissue sample. However, large tumors often accompany neovascularization, which increases the risk of bleeding. In addition, cases where biopsy is technically difficult due to the location of the target lesion or is difficult in pediatric patients and patients who cannot stay at rest, always concern healthcare professionals.

A survey on biopsy-related complications in 57 clinical studies has reported that the overall incidence of complications was 5.2% (39 of 745 subjects), with the incidence of major complications being 0.8% (6 of 745 subjects). Especially, the incidence of complications from intrathoracic biopsy was high, at 17.1% (36 of 211 subjects). These cases included five Grade 3 events, requiring therapeutic interventions [14]. At present, various clinical studies are in progress, and many of them require biopsy for accurate evaluation. In these studies, if biopsy can be replaced with ctDNA assay, the burden on patients will be greatly reduced.

## 2. Measurement of ctDNA

The proportion of tumor-derived DNA fragments in blood is usually very small: ctDNA is considered to account for less than 1% of circulating plasma DNA [15]. This fact has made it difficult to use ctDNA as a diagnostic material in clinical practice in the early days following the discovery of ctDNA because of the lack of technologies to detect ctDNA. However, in recent years, the improvements in DNA analysis technologies, such as digital PCR and next-generation sequencing (NGS), have made it possible to detect trace amounts of DNA. Digital PCR has enabled accurate quantitative analysis, and NGS has enabled the comprehensive analysis of gene mutations.

Guardant360 CDx and FoundationOne Liquid CDx, which were approved by the U.S. Food and Drug Administration (FDA) in 2020, are multi-gene panel tests to detect cancer-related genes using ctDNA in solid tumors. In addition, Signatera, an assay based on highly sensitive analytical technologies that uses NGS, is a test that allows for generating multi-gene panels personalized for patients from their tissue specimens. Currently, a clinical study is being conducted to evaluate the risk of recurrence by detecting cancer-specific genetic mutations in the blood using Signatera in patients with radically resected colorectal cancer [16]. Thus, ctDNA also allows the detection of molecular residual disease (MRD), which is another of its features, and various clinical studies are being conducted.

## 3. Usefulness of ctDNA in Oncology

Liquid biopsy is expected to be useful in various situations of cancer care. We have reviewed the usefulness of ctDNA in each process of diagnosis, treatment, and recurrence (Figure 1). Table S1 summarize the examples of studies for breast cancer employing ctDNA.

### 3.1. Cancer Screening

In cancer screening, gastrointestinal endoscopy, X-rays, etc. are performed for the purpose of the early detection of cancer, and tumor markers are used for evaluation, although this is controversial. A low tumor burden in early-stage cancer naturally makes it difficult to detect ctDNA, and there is no established screening method for early diagnosis using ctDNA.

The Guardant360 CDx and FoundationOne Liquid CDx mentioned above are used for advanced solid tumors and cannot be used in the screening for early detection of cancer. However, early detection is important in reducing cancer mortality. There are many types of cancer that can be cured if local treatment is feasible before the development of distant metastases. In the field of liquid biopsy, therefore, many studies on its use for screening are ongoing. Among them, a study group from Johns Hopkins University School of Medicine succeeded in developing the test method dedicated to screening that enables the detection of eight types of cancers [17]. The test CancerSEEK, in which protein biomarkers and genetic biomarkers are combined, allows diagnosis with blood specimens. The original study was conducted on 1005 patients with eight types of cancer, including Stages I–III ovarian cancer, liver cancer, and breast cancer, without distant metastases. The results showed that the sensitivity was 70%, although it varied depending on cancer type. Examined by stage, the detectability at Stage II was 73%. Thus, cancer detection at a stage where surgical resection is available is very useful also for screening. In addition, it is noteworthy that a specificity of 99% was achieved with a very low false positive rate, and that the primary tumor site was identified in 83% of the patients. However, the study was conducted in patients already diagnosed as having cancer manifesting subjective symptoms, rather than with healthy individuals. Therefore, it is considered necessary to perform a study to confirm the cancer detection rate in the screening in asymptomatic individuals.

Other major cancer screening tests currently under study include GRAIL, PanSeer, LUNAR-2 and DELFI. Each of these tests will be briefly summarized (Table 1).

The GRAIL and PanSeer, LUNAR-2 assays enable early cancer diagnosis by analyzing DNA methylation. The results of a verification test with the GRAIL in cancer and non-cancer patients showed that the specificity was 99% and the sensitivity at Stages I–III was 67%, and that the assay also allowed the localization of the primary site [18]. Furthermore, the PanSeer assay analyzes DNA methylation in the plasma of non-cancer patients without symptoms. As a result, the assay enabled the identification of non-cancer patients who would be diagnosed as having cancer later on nearly four years earlier than the conventional diagnostic methods [19].

**Table 1.** Summary of ctDNA tests for cancer screening.

| Assay | Detection Method | Detectable Cancer Types |
| --- | --- | --- |
| | | (Testing Phase) |
| Cancer SEEK | simultaneousy evaluates the concentration of 8 cancer proteins and the presence of oncogene mutations. | 8 |
| GRAIL | | Over 50 |
| PanSeer | detects changes in DNA methylation | 5 |
| LUNAR-2 | | 1 (early-stage colorectal cancer) |
| DELFI | detects altered DNA fragment patterns | 7 |

LUNAR-2 has been developed for screening of an early-stage colorectal cancer. As an invasive examination, such as colonoscopy, is necessary to detect an early-stage colorectal cancer, people may become reluctant to receive health checkups. LUNAR-2 is currently being tested in a clinical trial with the aim of improving early detection rates and increasing the number of people receiving health checkups [20].

DELFI is a test method that focuses on differences in the DNA fragment size between healthy individuals and cancer patients. Differences in fragmentation are detected using artificial intelligence. The results of verification in 208 cancer patients and 215 non-cancer patients showed that the overall sensitivity was 73% and the specificity was 98%. By adding information on ctDNA mutations to the fragment size, the sensitivity increased to 91% [21].

As described above, various assays are currently being studied for the early diagnosis of cancer, and we hope to see the results in actual clinical practice in the future.

*3.2. Comprehensive Cancer Genomic Profiling*

Compared with the comprehensive genetic analysis using tissue specimens, liquid biopsy is advantageous in terms of not only its minimal invasiveness, but also the reduction in the time taken to obtain the result. Nakamura, Y., et al. reported a study evaluating the usefulness of genotyping using ctDNA in advanced solid tumors. The use of ctDNA genotyping significantly shortened the screening duration (11 days vs. 33 days, $p < 0.0001$) and improved the study enrollment rate (9.5% vs. 4.1%, $p < 0.0001$), compared with the use of tissue genotyping for patient enrollment in the study. There were no differences in the objective response rate (ORR) anord progression-free survival (PFS) between the two groups [22].

In addition, the use of ctDNA may be advantageous over tissue biopsy in terms of diagnostic accuracy. In tumor tissue biopsy, heterogeneity among tumors may occur in diagnosis, depending on the tumor puncture site or in the case of multiple tumors. In 2012, Swanton, C., et al. published a study in which DNA sequencing was performed to detect mutations in renal cell carcinoma and associated metastatic sites. The study reported that different mutations were detected depending on the tumor puncture site, even within a single tumor. In addition, there were differences in gene mutations even in specimens collected from metastatic sites in the same patient [23]. As a molecular-targeted drug is sometimes selected based on the detected DNA mutations, these differences in detected DNA mutations depending on the puncture site may affect patient prognosis. However, ctDNA is less susceptible to heterogeneity because it enables the measurement of the entire DNA derived from tumors in the whole body.

Cancer of unknown primary represents, as the name suggests, a group of malignant tumors whose primary lesion is difficult to identify because multiple lesions are present throughout the body at the time of detection. There is no standard treatment showing clear efficacy. At present, chemotherapy combined with platinum agents is performed, but the prognosis is often poor. Thus, established treatments for cancer of unknown primary are limited, and the patient's general condition is poor due to metastases to various organs at the time of detection. Therefore, protracted identification of the primary lesion should be avoided.

In a study reported by Kurzrock, R., et al., ctDNA samples were collected from 442 patients with cancer of unknown primary, and 90% of these patients were found to have specific mutations of targetable genes, such as TP53 and KRAS [24]. These findings indicate that ctDNA is evaluable in cancers of unknown primary origin, and the analysis of these mutations showed the possibility of treatment with molecular-targeted drugs.

### 3.3. In the Process of Assessing Prognosis and Risk of Recurrence

Currently, in many solid tumors, it is a common practice to perform neoadjuvant and adjuvant chemotherapy by evaluating the recurrence risk for each patient. For example, in the case of breast cancer, the recurrence risk is assessed, based on the subtype and tumor size based on pathological diagnosis from biopsy, as well as the number of axillary lymph node metastases. For patients determined to be at a high risk, neoadjuvant and adjuvant chemotherapy is performed. As the rationale for these regimens, whether the risk of recurrence is high is determined based on the vast amount of data collected from patients so far. It is considered that adjuvant therapy has reduced postoperative recurrence in many patients. However, chemotherapy imposes a physical burden on disease-free patients. Therefore, if reliable information on recurrence risk is obtained using ctDNA, the patient burden may be reduced.

Patients who are ctDNA-positive before surgery have a significantly poor postoperative prognosis. Therefore, the presence of ctDNA may be an independent prognostic factor, which can provide useful information on postoperative follow-up and the selection of adjuvant therapy.

Of patients with Stage II–III breast cancer who received neoadjuvant therapy, those in whom methylated DNA (met-DNA) was detected had significantly poor OS subsequently [25]. In addition, there is a study reporting that metastatic colorectal cancer patients with relatively low ctDNA concentrations had a significantly better prognosis than those with high ctDNA concentrations, showing a marked correlation between survival and ctDNA concentration [26].

### 3.4. Evaluation of Treatment Response and Drug Resistance

Generally, imaging evaluation and tumor marker levels are often used as measures to determine the effectiveness of chemotherapy, but these measures may be replaced with ctDNA. One study reported that when ctDNA and imaging evaluation were used to evaluate treatment response in patients with metastatic non-small cell lung cancer (NSCLC) receiving immune checkpoint inhibitors, changes in these two measures were correlated [27]. In addition, changes in ctDNA levels and subsequent survival were also correlated. Moreover, ctDNA reflects tumor burden in real time because of its short half-life, being from 16 min to 2.5 h, which makes ctDNA useful [28].

Furthermore, ctDNA is also useful for the reassessment of patients who were found to have progressive disease (PD) due to the development of drug resistance during the course of treatment. Although the mechanism of drug resistance is still unclear, it is known that ALK fusion proteins on the surface of cancer cells undergo secondary genetic mutation, which is thought to prevent the binding of ALK tyrosine kinase inhibitors (ALK-TKIs) and reactivate the growth of cancer cells [29].

Molecular-targeted drugs are effective for patients with ALK fusion gene-positive NSCLC. However, after several years of treatment with the molecular-targeted drugs, those patients sometimes develop resistance, resulting in PD. In that case, to evaluate the secondary mutations, comprehensive genetic analysis is required for drug selection. However, re-biopsy is sometimes difficult depending on the location of the PD lesion. According to one report about an analysis of genetic mutations in patients with metastatic NSCLC in 2019, non-inferiority of plasma specimens was demonstrated based on the concordance rate of plasma specimens against tissue specimens obtained in assays using plasma specimens [30]. Such results suggest that genetic testing using plasma specimens allows appropriate evaluation and drug selection, in some cases. The International Associ-

ation for the Study of Lung Cancer (IASLC) also recommends the use of plasma specimens in the re-evaluation of PD during molecular-targeted drug therapy for patients in whom re-biopsy is difficult [31].

Another example is the detection of acquired resistant mutation in breast cancer patients treated with the cyclin-dependent 4/6 inhibitors palbociclib and letrozole. Post treatment ctDNA showed several mutations considered to be resistant to palbociclib and letrozole [32]. For example some of the patients obtained THE ESR1 mutation, which is considered to be resistant to letrozole.

### 3.5. Early Detection of Recurrence

Next, we discuss the possibility that ctDNA can be a measure of recurrence.

In a study comparing imaging examinations with the assays of ctDNA, a tumor marker (CA15-3), and circulating tumor cells in 30 patients with metastatic breast cancer, ctDNA levels showed a greater dynamic range and greater correlation with changes in the tumor burden than CA15-3 or circulating tumor cells. Of the measures tested, ctDNA was the measure that most rapidly reflected the treatment response in 10 of 19 women (53%) [33].

Another study investigated whether ctDNA can be used for monitoring the molecular recurrence of residual disease (MRD) in breast cancer [34]. Using plasma ctDNA collected after surgery from patients with early-stage breast cancer receiving neoadjuvant chemotherapy and curative resection, MRD was quantitatively monitored. The results showed that, in a cohort of patients in whom MRD was determined by the detection of ctDNA in postoperative samples, ctDNA levels increased about two months before clinical recurrence. At the time of clinical recurrence, extensive lesions are often seen. Therefore, if recurrence is suggested by monitoring ctDNA levels, earlier therapeutic interventions will become feasible.

In addition, methylated ctDNA (met-ctDNA) has higher sensitivity than tumor markers and is correlated with the treatment responsiveness and residual tumor burden after surgery. Therefore, its usefulness as a measure of recurrence is also expected [35]. In one study, DNA methylation of RASSF1A during adjuvant therapy after mastectomy was monitored to determine whether patients were responsive or resistant to the treatment [25].

A study reported that the detection of MRD using ctDNA was useful in evaluating the risk of postoperative recurrence in early-stage colorectal cancer. Since ctDNA responds more quickly and sensitively than tumor markers, it is expected to be useful as a measure of the risk of postoperative recurrence in the future [36]. Thus, if ctDNA is established as a measure of postoperative recurrence, the frequency of imaging examinations at large hospitals may be reduced. In addition, if it becomes possible to follow up such patients at clinics, situations where patients are concentrated at large hospitals may be improved even a little.

## 4. Future Challenges with ctDNA in Clinical Practice

We have described various usefulness of ctDNA so far. In particular, compared to tissue biopsy, liquid biopsy using ctDNA is less invasive and easier to perform regardless of the target organ, which is a great advantage for both patients and medical professionals (Table 2). However, few ctDNA results have yet been established as diagnostic methods for treatment selection. The following issues need to be resolved before they can be used routinely in actual clinical practice.

First, it is still unclear to what extent tumor tissue is reflected in the information obtained from ctDNA. For example, with advancing age, clonal hematopoiesis is likely to take place even under normal conditions. This is called "clonal hematopoiesis of indeterminate potential" (CHIP). The CHIP might be assessed as abnormal by liquid biopsy, giving a false-positive result in elderly patients. As mentioned above, cfDNA increases by inflammatory diseases (such as infection and rheumatism) and hepatic function disorder, or after exercise and during pregnancy [10,37–39]. It appears that many clinical studies to date have selected patients excluding these conditions. However, the use of ctDNA

in actual clinical practice is likely to cause many false-positive results because the target patients are those with various medical histories and constitutions.

**Table 2.** Advantages and disadvantages of liquid biopsy and tissue biopsy.

| | Liquid Biopsy | Tissue Biopsy |
|---|---|---|
| advantages | <ul><li>less painful and minimally invasive</li><li>collected in a short time</li><li>can be collected regardless of the location of the target organ</li><li>can be re-taken multiple times</li><li>can reflect the tumor volume in real time</li></ul> | <ul><li>technology is established.</li><li>results can be useful for treatment selection</li></ul> |
| disadvantages | <ul><li>if the amount of ctDNA is insufficient, accurate information may not be obtained</li><li>analysis technology needs to be established</li><li>results may be influenced by the patient's condition or underlying disease</li></ul> | <ul><li>highly invasive</li><li>long time may be required for collection</li><li>difficult to collect depending on the target organ</li><li>cannot be easily re-taken</li></ul> |

Although DNA analysis technologies have been advancing, accurate detection may not be possible in the case of early-stage cancer with a low tumor burden or in the case of tumors localized to the central nervous system [26]. In addition, there are still issues, such as that differences in the ctDNA positive rate depending on the cancer type and recurrence pattern, the optimal analysis timing, and positivity assessment criteria have not yet been established.

Moreover, there is the possibility that tumor-derived ctDNA may be diluted by a relative increase in cfDNA derived from leukocytes due to leukocytolysis during the blood coagulation process. Therefore, it is considered ideal to complete the centrifugation and analysis processes within 6 h of blood drawing [40,41]. However, there are some issues here, too, such as that a system equipped with plasma centrifugation and analysis technologies, which are required to obtain accurate results, is difficult to establish at small-scale institutions. Therefore, further studies are required for the use of ctDNA in clinical practice at many institutions. Such studies include the development of blood collection tubes that prevent the coagulation and deterioration of specimens for a long period of time.

Oncology deals with complicated situations: onset patterns, mode of advancement, and prognosis greatly change depending on the tumor location, affected organ, the patient's age, race, sex, and comorbidities. Therefore, whether and to what extent liquid biopsy can be applied as a screening technology to a wide variety of patients uniformly is a major issue to be addressed. Currently, the effectiveness of liquid biopsy is seen in every aspect of medical care, and the development of the related technologies is being promoted, which is promising.

**Supplementary Materials:** The following are available online at https://www.mdpi.com/article/10.3390/pr9122198/s1, Table S1. Some of the current or upcoming studies using ctDNA (cited from ClinicalTrials.gov).

**Author Contributions:** Conceptualization, S.S. and Y.N.; methodology, S.S. and Y.N.; data curation, S.S. and Y.N.; writing—original draft preparation, S.S. and Y.N.; writing—review and editing, S.S. and Y.N. All authors have read and agreed to the published version of the manuscript.

**Funding:** This research received no external funding.

**Institutional Review Board Statement:** Not applicable.

**Informed Consent Statement:** Not applicable.

**Data Availability Statement:** Not applicable.

**Conflicts of Interest:** S.S. declare no conflict of interest. Y.N.: speaker's bureau from AstraZeneca, Eisai, Ono, Gardant, Takeda, Eli Lilly, Novartis, Pfizer, Chugai, Fuji Film Toyama Chemistry, Taiho, Mundi, Bristol, Shionogi.

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
