# Peer review of "Circulating Tumor DNA in Oncology"

_processes, doi:10.3390/pr9122198_

Round 1

Reviewer 1 Report

Overall, this paper has potential but at the moment has too many issues to recommend acceptance. 

The paper is a little disjointed - it jumps around topics, rather than providing a clear overview of the field. A review should be accessible to a scientist not in the field, and I don't believe this review does that. It focusses too much on individual papers rather than presenting an overview or synthesising conclusions. 

The table at the end is a good start. It needs to be commented on, rather than presented blindly at the end. I would incorporate far more of the table into the text and reorganise around that. 

In addition, some specific points for correction:

Line 17: References should not appear in the abstract.

  • The abstract itself focusses too much on background information, and not enough on what this paper adds. It reads like an introduction.

Line 50: 'ctDNA is considered to account for less than 1% of circulating plasma' : this requires a reference

Line 97: change 'healthy individuals' to 'asymptomatic individuals'

Line 98: If 'Gardant LUNAR-2' is mentioned, it needs to be discussed in the text along with the other technologies. 

Line 115: 'future results are expected' is too vague. Can you be more specific?

Line 118: Details of the negative aspects of traditional biopsies should be introduced in the introduction. 

Section 3.4: Can you add details of pharmacogenetic uses of ctDNA? 

Overall, this article has excellent potential and clearly a lot of work has gone into it. It just needs to be a higher-level overview of the field, rather than a list of technologies. 

Author Response

The paper is a little disjointed - it jumps around topics, rather than providing a clear overview of the field. A review should be accessible to a scientist not in the field, and I don't believe this review does that. It focusses too much on individual papers rather than presenting an overview or synthesising conclusions. 

The table at the end is a good start. It needs to be commented on, rather than presented blindly at the end. I would incorporate far more of the table into the text and reorganise around that. 

Response: Thank you for your review. I agree the reviewer. However in consideration of second reviewer who recommended to add figure, we added figure for better understanding and add reference in introduction.

We added 

"Liquid biopsy is broadly defined as a minimally invasive examination to detect ctDNA (Figure), circulating tumor cells (CTC), miRNA, etc. in body fluids [5–7]."

in page 3 line 3.

Line 17: References should not appear in the abstract.

Response: Thank you for your suggestion. We deleted reference in the abstract.

The abstract itself focusses too much on background information, and not enough on what this paper adds. It reads like an introduction.

Response: Thank you for your suggestion. We added 

"In addition, the prognosis of cancer varies considerably depending on the degree of progression at the time of detection. Therefore, early detection of cancer is of utmost importance. With the increase in health consciousness, more people undergo regular health checkups, and it becomes necessary to diagnose cancer in a larger number of patients at an earlier stage. Although the accuracy of early detection has been improved by new imaging tests and examination techniques, each organ must be examined separately, and some organs are more difficult to examine than others in a regular health checkup.

The process of cancer screening, diagnosis, and detection of recurrence after treatment is extensive. It can also be expensive, and some of the examinations may be invasive. If all of these processes can be replaced by the analysis of ctDNA in liquid biopsy, only a single blood sample is required." 

in the abstract body (P2 L5).

Line 50: 'ctDNA is considered to account for less than 1% of circulating plasma' : this requires a reference

Response: Thank you for your suggestion. We added reference ([14]).

Line 97: change 'healthy individuals' to 'asymptomatic individuals'

Response: Thank you for your suggestion. We corrected.

Line 98: If 'Gardant LUNAR-2' is mentioned, it needs to be discussed in the text along with the other technologies. 

Response: Thank you for your suggestion. We added discussion.

"LUNAR-2 has been developed for screening of an early stage colorectal cancer. Because an invasive examination such as colonoscopy is necessary to detect an early stage colorectal cancer, people may become reluctant to receive health checkups. LUNAR-2 is currently being tested in a clinical trial with the aim of improving early detection rates and increasing the number of people receiving health checkups.[19]" was added in page 6 line 3.

Line 115: 'future results are expected' is too vague. Can you be more specific?

Response: Thank you for your suggestions. We modified the sentences to "As described above, various assays are currently being studied for the early diagnosis of cancer, and we hope to see the results in actual clinical practice in the future."

Line 118: Details of the negative aspects of traditional biopsies should be introduced in the introduction. 

Response: Thank you for your suggestion. We added "The larger the tumor, the easier it is to collect a tissue sample. However, large tumors often accompany neovascularization which increases the risk of bleeding." in page line 20.

Section 3.4: Can you add details of pharmacogenetic uses of ctDNA? 

Response: Thank you for your suggestion. 

We added "Although the mechanism of drug resistance is still unclear, it is known that ALK fusion proteins on the surface of cancer cells undergo secondary genetic mutation which thought to prevent the binding of ALK tyrosine kinase inhibitors (ALK-TKIs) and reactivate the growth of cancer cells.[29] " in page 10 line 3.

Overall, this article has excellent potential and clearly a lot of work has gone into it. It just needs to be a higher-level overview of the field, rather than a list of technologies. 

The comments offered by the reviewer have been helpful in formulating what we believe is a stronger paper. We appreciate these thoughtful comments.

Reviewer 2 Report

The attached manuscript is very interesting. It provides some new information about the importance of Circulating DNA in the diagnosis of cancers. 

However, it is highly recommended to add some figures and charts which is crucial in certain reviews.

Author Response

The attached manuscript is very interesting. It provides some new information about the importance of Circulating DNA in the diagnosis of cancers. 

However, it is highly recommended to add some figures and charts which is crucial in certain reviews.

Response: Thank you for your suggestion. We added figure and mentioned in the text (page 3 line 4).

The comments offered by the reviewer have been helpful in formulating what we believe is a stronger paper. We appreciate these thoughtful comments.

Round 2

Reviewer 1 Report

Thank you for addressing our comments so far. Unfortunately this version did not have line numbers. 

  • The figure is a useful addition to this paper. However, it requires a figure legend and clear references to the figure in the text - i.e. Figure 1, not just 'figure'. 
  • Another useful addition would be a table comparing different technologies (GRAIL, PanSeer, etc). 
  • I would also recommend a table of the benefits of ctDNA/cfDNA compared to traditional biopsy
  • Appreciate the additional reference to risks in traditional biopsy being added. However, I think the idea should be introduced even earlier, in the introduction. Perhaps after the sentence 'Liquid biopsy is broadly defined as a minimally invasive examination to detect ctDNA, circulating tumor cells (CTC), miRNA, etc. in body fluids.'
  • That sentence needs further elaboration, to show that this article is only focussing on ctDNA. I would not mention CTC or miRNA as they do not come up later in the article. 
  • Appreciate the reference to ALK being added. A further (non-ALK) example of pharmacogenetic applications in ctDNA would be useful. 
  • Again, the table at the end of the paper adds little without extensive discussion in the text. I would advise that the table is instead supplied as supplementary material. Alternatively, it would need to be integrated into the text and a conclusion should then appear after the table. A correct legend also needs to be supplied, including abbreviations. 

Author Response

  • The figure is a useful addition to this paper. However, it requires a figure legend and clear references to the figure in the text - i.e. Figure 1, not just 'figure'. 

Thank you for your suggestion.

We added "

Liquid biopsy is broadly defined as a minimally invasive examination to detect ctDNA (Figure1), circulating tumor cells (CTC), miRNA, etc. in body fluids [11–13]. We will focus on ctDNA in this review.

Currently, a biopsy of the primary or metastatic lesion is performed in cancer diagnosis, and a definitive diagnosis is made using pathological diagnosis. The process up to the diagnosis involves frequent medical exposure during diagnostic imaging and patients' physical distress associated with biopsy. The larger the tumor, the easier it is to collect a tissue sample. However, large tumors often accompany neovascularization which increases the risk of bleeding. In addition, cases where biopsy is technically difficult due to the location of the target lesion or is difficult in pediatric patients and patients who cannot stay at rest always concern healthcare professionals.

A survey on biopsy-related complications in 57 clinical studies has reported that the overall incidence of complications was 5.2% (39 of 745 subjects), with the incidence of major complications being 0.8% (6 of 745 subjects). Especially, the incidence of complications from intrathoracic biopsy was high, being 17.1% (36 of 211 subjects). These cases included five Grade 3 events, requiring therapeutic interventions [14]. At present, various clinical studies are in progress, and many of them require biopsy for accurate evaluation. In these studies, if biopsy can be replaced with ctDNA assay, the burden on patients will be greatly reduced."

in the introduction (page 3 line 13) and referred as "Figure 1".

Figure legend was added at the end of manuscript.

  • Another useful addition would be a table comparing different technologies (GRAIL, PanSeer, etc). 

Thank you for your suggestion. We added Table 1 and referred as "Each of these tests will be briefly summarized (Table1)." in page 6 line 9.

  • I would also recommend a table of the benefits of ctDNA/cfDNA compared to traditional biopsy

Thank you for your suggestion. We added Table 2 and referred as "

We have described various usefulness of ctDNA so far. In particular, compared to tissue biopsy, liquid biopsy using ctDNA is less invasive and easier to perform regardless of the target organ, which is a great advantage for both patients and medical professionals (Table 2). However, few ctDNA results have yet been established as diagnostic methods for treatment selection. The following issues need to be resolved before they can be used routinely in actual clinical practice."

in page 12 line 4.

  • Appreciate the additional reference to risks in traditional biopsy being added. However, I think the idea should be introduced even earlier, in the introduction. Perhaps after the sentence 'Liquid biopsy is broadly defined as a minimally invasive examination to detect ctDNA, circulating tumor cells (CTC), miRNA, etc. in body fluids.'

Thank you for your suggestion. We added "

Liquid biopsy is broadly defined as a minimally invasive examination to detect ctDNA (Figure1), circulating tumor cells (CTC), miRNA, etc. in body fluids [11–13].We will focus on ctDNA in this review.

Currently, a biopsy of the primary or metastatic lesion is performed in cancer diagnosis, and a definitive diagnosis is made using pathological diagnosis. The process up to the diagnosis involves frequent medical exposure during diagnostic imaging and patients' physical distress associated with biopsy. The larger the tumor, the easier it is to collect a tissue sample. However, large tumors often accompany neovascularization which increases the risk of bleeding. In addition, cases where biopsy is technically difficult due to the location of the target lesion or is difficult in pediatric patients and patients who cannot stay at rest always concern healthcare professionals."

in the introduction (page 3 line 13).

  • That sentence needs further elaboration, to show that this article is only focussing on ctDNA. I would not mention CTC or miRNA as they do not come up later in the article.

Thank you for your suggestion. We added " We will focus on ctDNA in this review." to clarify.

  • Appreciate the reference to ALK being added. A further (non-ALK) example of pharmacogenetic applications in ctDNA would be useful. 

Thank you for your suggestion. We added "Another example is the detection of acquired resistant mutation in breast cancer patients treated with cyclin-dependent 4/6 inhibitor palbociclib and letrozole. Post treatment ctDNA showed several mutations considered to be resistant to palbociclib and letrozole [32]. For example some of the patients obtained ESR1 mutation, which is considered to be resistant to letrozole." in page 10 line 19.

  • Again, the table at the end of the paper adds little without extensive discussion in the text. I would advise that the table is instead supplied as supplementary material. Alternatively, it would need to be integrated into the text and a conclusion should then appear after the table. A correct legend also needs to be supplied, including abbreviations. 

Thank you for your suggestion. We changed the Table to supplemental material Table S1.
